# Exploring collective emotion transmission in face-to-face interactions

**Wen Zheng** [1,2☯*], **Ailin Yu**[1], **Ping Fang**[3☯], **Kaiping Peng**[1*]

**1** Department of Psychology, Weiqing Building, Tsinghua University, Beijing, China, **2** Department of Medical Psychology, Capital Medical University, Beijing, China, **3** School of Psychology, Capital Normal University, Beijing, China

☯ These authors contributed equally to this work.
* xiaofeixia_z@163.com (WZ); pengkp@mail.tsinghua.edu.cn (KP)

**Data Availability Statement:** The datasets generated and analysed during the current study are available on the Open Science Framework (osf. io/xhjyf/).

**Funding:** National Social Science Foundation Grant number: BBA160046.

## Abstract

Collective emotion is the synchronous convergence of an effective response across individuals toward a specific event or object. Previous studies have focused on the transmission of cyber collective emotion; however, little attention has been paid to the transmission of collective emotion in face-to-face interactions. Using an experimental design, we examined how emotions are transmitted from some members to the whole group in face-to-face situations. We used a news report of a social event as an emotion stimulus to induce anger and disgust in 158 middle school students aged 12 to 15, with an average age of 13.20 years ($SD = 0.651$) We randomly assigned one-third of the participants to be "transmitters," while the others were "receivers." Transmitters shared their feelings with receivers; then, receivers communicated with other group members. The results indicated that negative collective emotions were transmitted from high- to low-intensity members, which converged through the effect of emotional contagion. It accumulated through the effect of an emotional circle, during which the feedback reinforced emotion intensity. The collective emotion transmission model comprised emotion diffusion, contagion, and accumulation. This model elucidates the intrinsic features of collective emotion transmission, enriches the research on collective emotion, and provides theoretical references for monitoring and managing future public events.

## Introduction

Human emotions are individual, one-way, and unrepeatable phenomena [1,2]. Researchers have increasingly realized that emotions at the collective level play a key role in our daily lives. Collective emotion is the synchronous convergence of an effective response across individuals toward a specific event or object [2,3]. Collective emotions constitute a wide range, such as global panic concerning the coronavirus or the public's excitement after their country's win at the Olympics. Ample research has examined cyber collective emotions [4,5]; however, studies on collective emotional transmission in face-to-face situations mainly focus on the dyad interactions [6,7]. A face-to-face situation refers to a condition in which many people gather

**Competing interests:** The authors have declared that no competing interests exist.

together in the same spot: namely, the formation of an "offline" event. How do people transmit emotions at the collective level? Does this transmission in face-to-face interactions follow the same rules as those of cyber collective emotions? Exploring how collective emotion is transmitted in face-to-face interactions can provide empirical and theoretical support to the understanding of collective emotions and provide further guidance for addressing public events.

Researchers have focused on cyber collective emotions using computational simulation or big data [4–10]. The essence of collective emotion transmission is emotional information transmission among group members [11]. Emotion transmission follows the general pattern of information transmission; however, the special features of emotion make emotion transmission different from information transmission. Many studies have addressed the area of information transmission, wherein the epidemic model and the heat transfer model are the most commonly used [9]. However, few studies have considered the psychological process of how emotions are transmitted from one person to another based on the theory of emotion social sharing [12,13] and emotional contagion [20]. This study considered three aspects of emotion: diffusion, convergence, and accumulation.

Emotional arousal is the reason for sharing stories, news, and information. High emotional arousal strengthens an emotional experience and has high emotional intensity [14]. Emotions with the same valence that are transmitted with a higher level of intensity will cause broader transmission. When people fiercely express their emotions, they are easily noticed by others and have an increased level of exposure, which enables their emotions to be transmitted easily among group members [15]. Thus, emotions within a group are always transmitted from high-intensity members to low- intensity ones, in line with the heat transfer model [10].

In face-to-face situations, collective emotion is mainly transmitted by emotional contagion [5,16–18]. Individuals in a crowd will automatically imitate others' facial expressions, intonations, gestures, actions, and more to acquire the emotions of others because of the activation of mirror neurons [19]. The emotional contagion of a pair of individuals has been widely studied [20,21]; however, little attention has been paid to emotion transmission among group members, which is more complicated than transmission between two individuals.

When a person joins a group, he/she is influenced by other members' emotions. This process happens interactively among many group members [22]. Transmitters express their emotions via their expressions, voice, tones, and gestures [23]. In turn, receivers' emotional feedback affects transmitters' emotional state. Consequently, an emotion cycle is formed between transmitters and receivers [24]. This emotion cycle enables repetition and intensification of emotion within the group. The end result is that the emotion cycle drives the collective emotion to homogenization [25]. Therefore, we proposed the following hypotheses:

H1: When the homogeneity of collective emotion is low, collective emotion will be transmitted from members with strong negative emotions to members with weak negative emotions, and it will be gradually distributed.

H2-a: When negative collective emotion has low intensity and low homogenization, group members will achieve emotional convergence through emotional contagion.

H2-b: When negative collective emotion has low intensity and low homogenization, negative responses from others will induce collective emotion in oneself, which means group members will achieve emotional convergence through an emotion cycle among the group members.

H3: When negative collective emotion has high intensity and high homogenization, emotional contagion and an emotion cycle will not continuously strengthen the collective emotion; however, the emotion intensity will not be weakened.

In sum, we examined how emotions are transmitted from some members to the whole group in face-to-face situations. We induced negative emotions in groups to conduct an exploratory investigation of the emotional transmission of collective emotions. This exploration provides novel insights into the understanding of collective emotions.

## Materials and methods

### Participants

Eighty junior grade-one students and 78 junior grade-two students were randomly recruited from a middle school in Beijing, China; of these, 78 were boys and 80 were girls. Their ages ranged from 12 to 15 years, with a mean age of 13.20 years (*SD* = 0.651).

All participants provided written informed consent in accordance with the Declaration of Helsinki. This study was also conducted in accordance with the recommendations of the guidelines of the Human Research Ethics Committee of Capital Normal University, and written informed consent was obtained from all participants' parent/guardian. The protocol was approved by the Human Research Ethics Committee of Capital Normal University.

### Material

**Emotion stimuli.** The material was based on a real event—a public press release: "An Asian country's media insists that movable-type printing was invented by them." This collective event is not personally relevant; however, it had symbolic meaning for collective self-esteem. Researchers collected and synthesized relevant reports on this issue from national and international media. They then distilled these reports into a news item consisting of about 1,000 Chinese characters (S1 File).

**Emotion ratings.** Participants were presented with eight emotions (sad, happy, angry, disgusted, satisfied, surprised, excited, and calm) [26]. They were asked, "As a Chinese person, to what extent do you feel each of the following emotions?" They responded on seven-point scales, ranging from *not at all* to *very much*. Factor analyses revealed that *angry* and *disgusted* formed the negative emotion dimension (Cronbach's *α* of .804), while *happy*, *satisfied*, and *excited* formed the positive emotion dimension (Cronbach's *α* of .842).

**Emotional contagion ratings.** Participants were asked to rate the extent to which they perceived anger and disgust from transmitters' language, facial expressions, actions, and intonations when communicating the news material, from 1 (*not at all*) to 7 (*very much*). After the factor analysis, *angry* and *disgusted* were combined into one dimension of negative emotional contagion. The Cronbach's *α*s of language, facial expression, and actions ranged from .894 to .935.

**Transmission feedback ratings.** One item with two possible emotions was used to test participants' emotion perception of their counterparts when they transmitted emotions as a transmitter and when they received emotions as a receiver. They were asked, "When you expressed your emotions of this news to him/her, what was his/her emotional reaction?" They were asked to rate this perception from 1 (*not angry/disgusted at all*) to 7 (*very angry/disgusted*). The Cronbach's *α* of this item was .902.

### Procedure

Experiments were conducted in four groups, with 35–40 participants in each group. The procedure is shown in Fig 1. First, all participants evaluated their emotional baseline. The pilot study showed that, in classroom conditions, participants tended to transmit their emotions to 2.33 people (*SD* = 0.637), ranging from 0–8 people. The experimenter randomly assigned one-

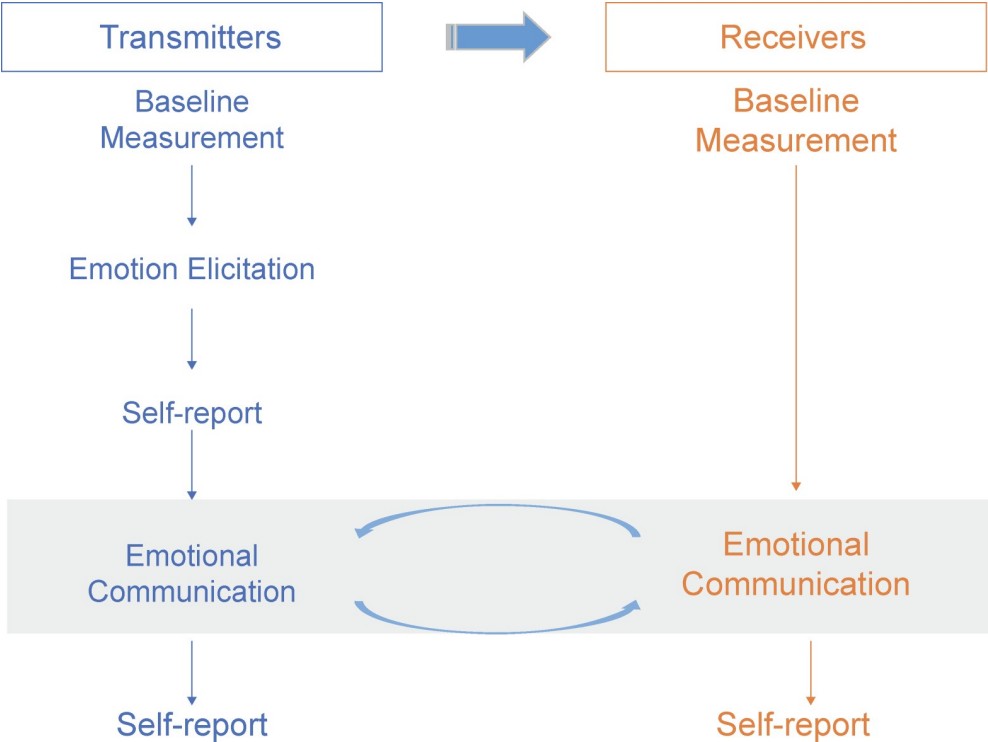

**Fig 1. Collective emotion transmission procedure of group members.**

third of participants to be transmitters. Then, the transmitters read the emotion induction material, while the other members of the group—the receivers—quietly waited. After reading, transmitters were asked to complete self-evaluations of their emotions. The experimenter informed the participants that he/she will leave the room; then, they would have 10 minutes to communicate with each other. The transmitters communicated both the information and their feelings about the reading material freely to the receivers for a maximum of 10 minutes. The receivers also expressed their opinions and feelings to the transmitters. Finally, the experimenter returned to the room, and all participants rated their emotions.

After the evaluation, participants were debriefed and their cooperation acknowledged with a compensation equivalent to 1.5 USD.

## Data analysis

There were 57 transmitters and 101 receivers. Because four receivers provided less than 50% of response items, we deleted their data. Thus, the data of 57 transmitters and 97 receivers were considered in further analyses. We conducted all the data analyses with SPSS 16.0 (IBM, Armonk, NY, USA).

The average of *angry* and *disgusted* ratings was computed as the indicator of negative emotion, and the average of *happy*, *excited*, and *satisfied* ratings was computed as the indicator of positive emotion. We compared the emotional changes in the receivers after receiving the emotion and the emotional convergence of the receivers after receiving the emotion using paired sample *t*-tests.

The collective emotional convergence of receivers before and after transmission needed to be compared. Previous studies mostly adopted *correlation coefficients* as an indicator of emotional convergence—where a positive correlation refers to convergence and a negative

correlation refers to divergence [27–29]. However, we adopted the *coefficient of variation* as the indicator of collective emotional convergence. The mean reflects the intensity of collective emotion, while the standard deviation reflects dispersion. Thus, the coefficient of variation (standard deviation/mean) can reflect collective emotional convergence—with a smaller coefficient indicating high homogeneity and, therefore, high convergence, and a larger coefficient indicating low homogeneity and, thus, low convergence [30]. We compared receivers' emotional baselines and their collective emotion after transmission. To determine the effect of the emotional contagion on collective emotion transmission, we tested group members' perceived emotional contagion of other members, using both verbal and nonverbal cues (language, facial expression, action, and intonation); then, we conducted correlation analyses with their emotion intensity changes after transmission.

# Results

## Manipulation check

The results of the paired sample *t*-tests indicated that negative emotion after inducement was significantly higher than at baseline ($t(56) = 13.453$, $p < .05$) and positive emotion after inducement was significantly lower than at baseline $(t(56) = 8.866$, $p < .05)$. The emotion levels at baseline, after inducement, and after transmission are shown in Table 1. Emotion-inducing material significantly induced the negative emotions of transmitters and reduced their positive emotions.

## Collective emotion transmission

**Emotion diffusion.**   The results showed that receivers' negative emotions after transmission were significantly more intense than at baseline ($t(96) = 11.947$, $p < .05$; Table 1). Transmission of negative emotion from transmitters to receivers significantly induced receivers' negative emotion. As shown in Fig 2, after transmission, receivers' positive emotion significantly decreased and negative emotion significantly increased.

Furthermore, the coefficient of variation of receivers' negative emotion after transmission was smaller than at baseline. As shown in Fig 3, after transmission, receivers' emotions indicated obvious convergence compared to baseline levels. This implies that emotion transmission promotes emotional convergence within group members.

In conclusion, following emotion transmission from transmitters to receivers, negative emotion flowed from high- to low-intensity members—enabling negative emotion diffusion in the whole group and leading to reaction convergence of group members.

**Table 1.  Participants' emotion levels at baseline, after inducement, and after transmission (N = 154).**

|  |  | Transmitters | | | Receivers | | |
|---|---|---|---|---|---|---|---|
|  |  | Mean | SD | COV | Mean | SD | COV |
| **Baseline** | **Negative emotion** | 2.105 | 1.546 | .734 | 1.799 | 1.357 | .754 |
|  | **Positive emotion** | 3.661 | 1.768 | .483 | 3.570 | 1.541 | .432 |
| **After inducement** | **Negative emotion** | 5.983 | 1.620 | .271 | - | - | - |
|  | **Positive emotion** | 1.310 | .757 | .578 | - | - | - |
| **After transmission** | **Negative emotion** | 5.600 | 1.752 | .313 | 4.933 | 2.236 | .453 |
|  | **Positive emotion** | 1.599 | 1.334 | .834 | 1.701 | 1.256 | .738 |

COV: coefficient of variation, SD: standard deviation.

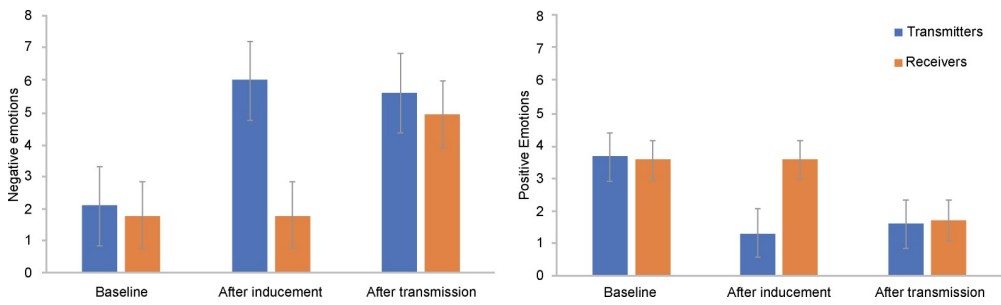

**Fig 2. Transmitters' and receivers' changes in emotion before and after transmission.**

**Influence of emotional contagion on emotion after transmission.** The results indicated that the participants' emotion intensity changes were significantly and positively correlated with their perception of their counterparts' language, actions, facial expressions, and intonations ($r_{language}$ = .472, $p < .01$, $r_{action}$ = .522, $p < .01$, $r_{facial\ expression}$ = .509, $p < .01$, and $r_{intonation}$ = .572, $p < .01$).

Taking receivers' negative emotion after transmission as the dependent variable, the transmitters' language, actions, facial expressions, and intonation contagion were used as predictor variables. The results showed that equation of transmitters' emotional contagion on receivers' negative emotion after transmission was significant ($R^2$ = .352, $F(4,78)$ = 10.587, $p < .001$). During transmission, language contagion negatively predicted emotion after transmission ($\beta$ = —.120, $p < .05$); action contagion positively predicted emotion after transmission ($\beta$ = .105, $p < .05$); facial expression contagion positively predicted emotion after transmission, but not significantly ($\beta$ = .07, $p > .05$); and intonation contagion significantly positively predicted emotion intensity after transmission ($\beta$ = .171, $p < .05$). To summarize, when transmitters transmit emotion to receivers, the stronger the emotion as expressed by the transmitters' intonation, the stronger the emotion for the receivers after transmission.

In contrast, emotion changes of transmitters after transmission were not significantly related to emotional contagion ($p > .05$). This means that, among transmitters, emotional contagion had no influence on their emotion changes after transmission.

**Influence of emotion feedback on emotion after transmission.** Participants' feedback was significantly positively correlated with receivers' emotion changes after transmission ($r$ = .525, $p < .01$), indicating that the stronger the emotion intensity of feedback, the stronger the negative emotion of receivers after transmission. However, such a correlation was not significant for transmitters ($p > .05$).

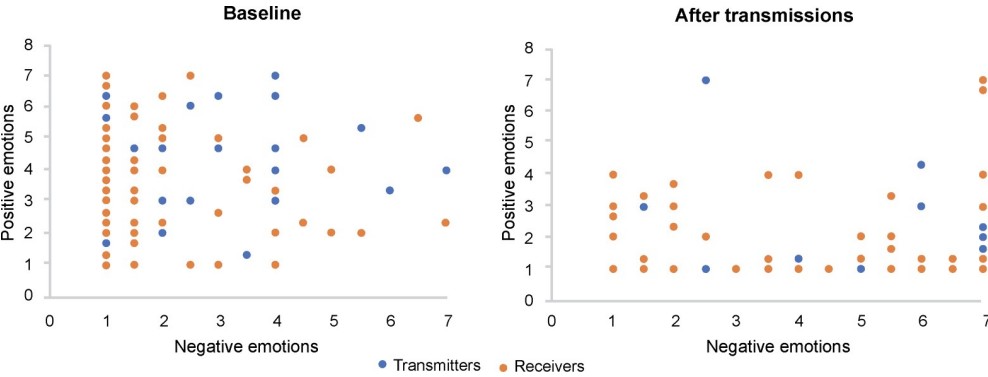

**Fig 3. Emotional convergence of transmitters and receivers before and after transmission.**

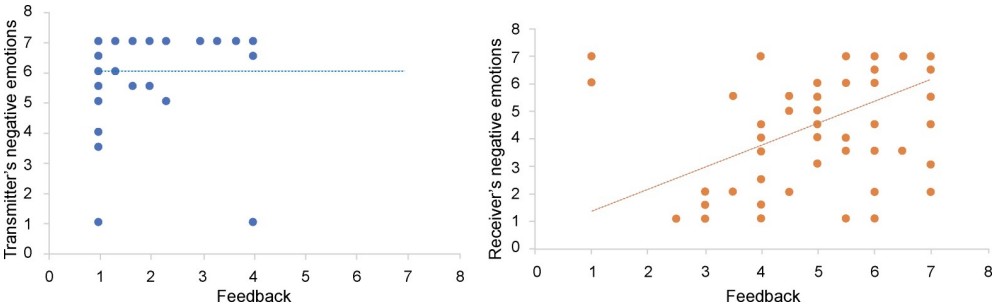

**Fig 4. Influence of emotion feedback on transmitters' and receivers' negative emotion after transmission.**

We developed a regression equation using participants' feedback as the predictor variable and receivers' emotion after transmission as the dependent variable. The results indicated that the equation of participants' negative feedback and negative emotions of receivers after transmission was significant ($R^2$ = .327, $F(1, 87)$ = 42.288, $p < .001$). Transmitters' negative emotional feedback significantly positively predicted the negative emotion of receivers after transmission ($\beta$ = .981, $p < .001$). This indicates that the more negatively the counterparts expressed their emotions, the stronger the negative emotion of the receivers after transmission (Fig 4).

## Discussion

This study created an offline, face-to-face situation of middle school students' collective emotion transmission and induced negative collective emotion to explore the collective emotion transmission model. The results showed that, after emotional induction, transmitters' negative emotions were significantly higher than at baseline and compared to those of receivers. After transmission, the negative emotions of receivers were significantly higher than at baseline. The emotional contagion of transmitters, including language, facial expressions, intonations, and actions, positively predicted receivers' emotional intensity. Transmitters' feedback positively predicted receivers' emotion intensity after transmission, indicating that stronger feedback was associated with more intense negative emotions among the receivers.

The results also revealed that in collective emotion transmission, emotion first flowed from high-intensity members (transmitters) to low-intensity members (receivers). In this procedure, transmitters' verbal and nonverbal emotional cues significantly influenced receivers' emotional intensity after transmission. The fiercer the participants' expression of their emotions, the more intense the negative emotions of the receivers. Emotion diffusion and contagion both promoted the negative emotion intensity in the group, which eventually led to convergence. Further, the more negative the feedback that participants received during transmission, the stronger the receivers' negative emotions. This indicates that emotion feedback played an emotion-strengthening role during this transmission.

### Collective emotion transmission model

After emotional induction, transmitters' negative emotions were significantly higher than at baseline and compared to those of receivers. After transmission, receivers' negative emotions were significantly higher than at baseline. These results support Hypothesis 1, indicating that collective emotion gradually transmits from high-intensity members (transmitters) to low-

intensity members (receivers). Although receivers' negative emotions are induced, the emotion intensity of transmitters shows no obvious change after transmission. Emotional energy flows from transmitters to receivers until the whole group's emotion converges.

Rime's social sharing theory of emotion supports our results [31]. When an intense emotional event affects a given individual, numerous members of this person's group are informed of it [32]. This emotion diffusion procedure is called the *flow effect*. Collective emotion transmission in offline situations not only demonstrates this flow effect, but emotional contagion also promotes emotion homogeneity.

The emotional contagion of transmitters, including language, facial expressions, intonations, and actions, positively predicted receivers' emotional intensity. This result supports Hypothesis 2. In addition to the direct verbal transmission from transmitters, nonverbal emotion information also significantly influenced receivers' emotional intensity. Transmitters use both verbal (speech and words) and nonverbal (facial expression, actions, intonations, etc.) means to transmit emotion [33]. People can be unconsciously influenced by others' nonverbal information [34] and express emotions similar to those of others [35]. This effect is especially obvious in face-to-face situations [26]. In conversations and in face-to-face interactions, people automatically and continuously mimic and synchronize their movements with the facial expressions, voices, postures, movements, and instrumental behaviors of others [36]. Thus, when both verbal and nonverbal means of expression are used, group members continuously observe and feel other members' emotions, which leads to collective emotional homogeneity.

People's emotional experiences are affected by others' feedback [25,37,38]. Our results indicated that transmitters' negative feedback positively predicted receivers' emotion intensity after transmission, which is consistent with Hypothesis 3. The more negative the emotion embodied in the feedback, the more negative the receivers' emotion after transmission. When receivers transmit negative emotion to others, the more negative the emotion in the feedback, the stronger the negative emotion of the receivers. This cycle drives receivers to express stronger negative emotion to others. This cycle also coincides with Lishner and colleagues' viewpoint of emotional contagion [39]. Under the influence of such interactions, group members' negative emotions are continuously transmitted within groups and strengthened repeatedly during this procedure, thus forming collective emotion with certain intensity [25,39].

During negative collective emotion transmission, only receivers were influenced by emotion flow, emotional contagion, and emotion cycle effects, and these three effects did not have a significant effect on transmitters. Transmitters' emotion intensity did not demonstrate a significant change after transmission, and there was no obvious change regarding emotional convergence. This result shows that, when these effects gradually fade and the collective emotion within a group has high intensity and homogeneity, the transmission of emotion will stop. When transmitters' emotion has high intensity and convergence, the effects of emotion flow and contagion are non-significant. In contrast, when receivers' emotion initially has low intensity and convergence, emotion flow and contagion can continuously influence emotional intensity and gradually lead to emotional convergence.

## Limitations and further research

First, this study adopted a self-designed questionnaire to measure the dependent variables. More approaches can be applied to verify our results, such as behavioral observations and coding emotion transmission. Second, some researchers believe that emotional convergence has two paths: emotion-contagion-based and perception-analysis-based [20]. This study only controlled for group members' cognition homogeneity, without controlling the emotional contagion influence during transmission. Further studies should consider both emotional and

cognitive factors to clarify collective emotion transmission: for example, the interaction effects of emotional contagion and group identification on emotion transmission. Third, we explored the transmission model in a cluster environment; however, transmission of collective emotion in real-life situations does not always happen in such an environment. When group members cannot interact continuously with other group members, the effect of emotional contagion will be weakened. Other transmission features and their effects will also change. Further studies should examine the actual situations in which collective emotions occur, such as cyber collective emotions.

## Conclusions

This study explored middle school students' negative collective emotion transmission models in face-to-face situations by creating an offline transmission environment. The results revealed that the negative collective emotion transmission model consisted of emotion diffusion, contagion, and accumulation. Negative emotion was transmitted from high- to low-intensity members. Collective emotion achieved homogeneity through emotional contagion and accumulated power from a continuous emotion cycle. With the strengthening effect of feedback, it finally took form of a collective emotion with certain behavioral drives. This model elucidates collective emotion transmission and enriches the research on collective emotion.

## Supporting information

**S1 File. Emotion assessments and pilot study.**
(DOCX)

**S1 Fig. Participants' mean emotional intensity among four emotion-eliciting materials.**
(TIF)

## Acknowledgments

The authors thank Ms. Xiru Liu and Ms. Anting Lai for their assistance with the experiments and valuable discussion.

## Author Contributions

**Conceptualization:** Ping Fang, Kaiping Peng.

**Data curation:** Wen Zheng.

**Formal analysis:** Wen Zheng.

**Resources:** Ping Fang, Kaiping Peng.

**Supervision:** Ping Fang, Kaiping Peng.

**Writing – original draft:** Wen Zheng.

**Writing – review & editing:** Ailin Yu.

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
