## [Decision Letter · Decision Letter 0]

22 Apr 2020

PONE-D-20-06106

Exploring collective emotion transmission in face-to-face interactions

PLOS ONE

Dear Dr. Zheng,

Thank you for submitting your manuscript to PLOS ONE. After careful consideration, we feel that it has merit but does not fully meet PLOS ONE’s publication criteria as it currently stands. Therefore, we invite you to submit a revised version of the manuscript that addresses the points raised during the review process.

We would appreciate receiving your revised manuscript by Jun 06 2020 11:59PM. To enhance the reproducibility of your results, we recommend that if applicable you deposit your laboratory protocols in protocols.io, where a protocol can be assigned its own identifier (DOI) such that it can be cited independently in the future. For instructions see: http://journals.plos.org/plosone/s/submission-guidelines#loc-laboratory-protocols

We look forward to receiving your revised manuscript.

Kind regards,

Zezhi Li, Ph.D., M.D.

Academic Editor

PLOS ONE

Journal Requirements:

2. Please provide additional details regarding participant consent. In the Methods section, please ensure that you have specified whether participants were informed as to the purposes of your study and whether they were briefed about the potential risks of taking part. If your study included minors, state whether you obtained consent from parents or guardians.

3. Please include additional information regarding the survey or questionnaire used in the study and ensure that you have provided sufficient details that others could replicate the analyses. For instance, if you developed a questionnaire as part of this study and it is not under a copyright license more restrictive than CC-BY, please include a copy, in both the original language and English, as Supporting Information. If possible, please also provide any other materials used in your study (e.g. the news report), alongside an English translation." Do not ping with follow up.

Reviewers' comments:

Reviewer's Responses to Questions

**Comments to the Author**

1. Is the manuscript technically sound, and do the data support the conclusions?

Reviewer #1: Partly

Reviewer #2: Yes

2. Has the statistical analysis been performed appropriately and rigorously? 

Reviewer #1: Yes

Reviewer #2: Yes

3. Have the authors made all data underlying the findings in their manuscript fully available?

Reviewer #1: No

Reviewer #2: Yes

4. Is the manuscript presented in an intelligible fashion and written in standard English?

Reviewer #1: Yes

Reviewer #2: Yes

5. Review Comments to the Author

Reviewer #1: This study explored the collective emotion transmission in face-to-face interactions. The results show that the collective emotion transmission model consisted of emotion diffusion, contagion, and accumulation. Collective emotion was transmitted from high arousal members to low arousal members. There were a few elements of the manuscript the derail from its overall clarity and impact. These concerns and others are detailed below. A revision of these sections would improve the quality of this study.

1. In Conclusions line 357-358, the authors claimed that the collective emotion was transmitted from high arousal members to low arousal members. What kind of collective emotions? Should be very specific, negative? However, I don’t see any data to show how the arousal level was measured and which subject, how many subjects have a high or low arousal?

2. In the Introduction, the authors have 3 hypotheses, but in the Discussion section the authors didn’t talk about how their results support or reject each of hypothesis. It would be beneficial to the reader if the authors can discuss each of their hypothesis based on their findings.

3. This study adopted the coefficient of variation as the indicator of collective emotion convergence. In line 193-194, The author claimed, the coefficient of variation can better reflect collective emotion convergence. Why the coefficient of variation can better reflect collective emotion convergence? Any refs to support this?

4. Regarding the experiment procedure, how many people do transmitter talk to? Are the transmitters award of or were told they were transmitters?

5. In Results section line 206-207, t (56), t (55). Why the df of the two paired sample t-test is different?

6. In line 133-134, the author claimed, “The pilot study showed that this material can induce negative collective emotion of the participants.” However, there is no data support this. It would be great if the authors can include their data from the pilot study to back their claim.

7. In line 179 after deleting the missing data, what do you mean deleting missing data? I suppose it should be excluding subject who has missing data. What kind of data were missing? How many?

8. Regarding the subject, are the subject middle school student or high school student? In the Abstract it says middle school, however in other places it says high school student. Need to be accurate and consistent.

9. The figure legend should put under nether each figure and it is better to put figure within the main text so that it is easier to read.

10. There are some text formatting issues in line 142-143 (Cronbach’sα). Please correct.

Reviewer #2: COMMENTS

This study aimed to explore how emotions are transmitted from some members to the whole group in a face-to-face environment. Through a social event, 158 middle school students were induced to feel anger and disgust. The present study randomly assigned 1/3 of them as senders and others as receivers. Transmitters shared their feelings with the receivers, and the receivers then communicate with other group members. The results show that negative collective emotions are transmitted through the flow from high arousal members to low arousal members. It is converged through the role of emotional contagion. It is accumulated through the role of an emotional cycle, and in the process, feedback strengthens the intensity of emotion. This study shows that the mode of collective emotional transmission consists of three parts: emotional diffusion, contagion and accumulation. This model helps to understand the inherent characteristics of collective emotion transmission, enriches the study of collective emotion, and provides a theoretical reference for the monitoring and management of group events in the future.

Although this study is interesting, the following issues need to be addressed before it is accepted and published.

1. There are some typos and Chinglish expression in the whole manuscript. Please ask native speaker to help modify the language. For example, in line 54, “Collective emotqqions” is a very obvious spelling mistake. In line 142, “with Cronbach’sαof .804” is also a typo. There are still a lot of problems like this, please revise them carefully. In line 211 to 212,“Negative and positive emotion mean, standard deviation, and coefficient of variance before and after collective emotion transmission” is a very typical Chinglish expression. There are many similar expressions in the text, please modify them one by one.

2. “Much research has been performed on cyber collective emotions.” There is a lack of references here. Please add the related references.

3. The structure of the introduction is a little loose. In this part, the present study puts forward relevant hypotheses based on previous studies. But the two hypotheses are separated by several paragraphs, and at the end of reading, you will forget the previous hypothesis. It is suggested that a paragraph should be added at the end of the introduction to explain the purpose of this study and the scientific problems to be solved, which will be clearer.

4. With regard to the collective emotion induction, I am wondering whether it is beyond their cognition to ask middle school students between the ages of 12 and 14 to rate events with complex political and historical backgrounds?

5. In the part of data analysis, what are the criteria and basis for deleting data? This paper does not elaborate on this in detail.

6. Negative emotions include only anger and disgust, while positive emotions include happiness, excitement and satisfaction. Do you consider the problem of quantity mismatch? In addition, are there corresponding criteria and references for dividing positive and negative emotions?

7. The discussion section is not enough to discuss in detail whether the results are consistent or inconsistent with previous studies, and the relevant explanations. After a brief summary of the research results, the study gives the collective emotion transmission model, and then detailed the current results, but did not explain the relationship between this model and the results of this study. What I am very curious about is whether the results of this study support this model, or do not support this model, or can it be modified? I hope to see the corresponding discussion in the revised draft.

6. PLOS authors have the option to publish the peer review history of their article (what does this mean?). If published, this will include your full peer review and any attached files.

Reviewer #1: No

Reviewer #2: No

---

## [Author Response · Author response to Decision Letter 0]

5 Jun 2020

Comments to the Author

1. Is the manuscript technically sound, and do the data support the conclusions?

Reviewer #1: Partly

Reviewer #2: Yes

2. Has the statistical analysis been performed appropriately and rigorously?

Reviewer #1: Yes

Reviewer #2: Yes

3. Have the authors made all data underlying the findings in their manuscript fully available?

Reviewer #1: No

Reviewer #2: Yes

4. Is the manuscript presented in an intelligible fashion and written in standard English?

Reviewer #1: Yes

Reviewer #2: Yes

5. Review Comments to the Author

Reviewer #1: This study explored the collective emotion transmission in face-to-face interactions. The results show that the collective emotion transmission model consisted of emotion diffusion, contagion, and accumulation. Collective emotion was transmitted from high arousal members to low arousal members. There were a few elements of the manuscript the derail from its overall clarity and impact. These concerns and others are detailed below. A revision of these sections would improve the quality of this study.

1. In Conclusions line 357-358, the authors claimed that the collective emotion was transmitted from high arousal members to low arousal members. What kind of collective emotions? Should be very specific, negative? However, I don’t see any data to show how the arousal level was measured and which subject, how many subjects have a high or low arousal?

Response: Thank you for your comment. We did not measure emotional arousal; rather, we measured emotional intensity. We have revised this throughout for additional clarity.

2. In the Introduction, the authors have 3 hypotheses, but in the Discussion section the authors didn’t talk about how their results support or reject each of hypothesis. It would be beneficial to the reader if the authors can discuss each of their hypothesis based on their findings.

Response: Thank you for your comment. We have revised the Discussion section accordingly; specifically, we included a summary of the results and the relationship between each result and our hypotheses (pages 15–17, lines 275–316).

3. This study adopted the coefficient of variation as the indicator of collective emotion convergence. In line 193-194, The author claimed, the coefficient of variation can better reflect collective emotion convergence. Why the coefficient of variation can better reflect collective emotion convergence? Any refs to support this?

Response: We appreciate your feedback. Collective emotion is the synchronous convergence of an emotion response across individuals toward a specific event or object. When group emotions arise, the emotional intensity of the whole group increases, and when group emotions spread, the homogeneity of the whole group rises. We measured both the intensity of negative group emotions using eight emotion items and the coefficient of variation as an indicator of emotional convergence. If two datasets have great dispersion, or if their scales differ, it is not appropriate to use standard deviation directly for comparison. In this case, the effect of measurement scale should be eliminated, which the coefficient of variation—the ratio of the standard deviation to the average of the original data—can accomplish. It is an absolute value that reflects the degree of discrete data. It is possible to directly compare the degree of emotion convergence of individuals in different groups. We have added a reference to support this point (Brewer et al., 2020; pages 9–10, lines 168–178).

4. Regarding the experiment procedure, how many people do transmitter talk to? Are the transmitters award of or were told they were transmitters?

Response: We measured the number of subjects that transmitters talked to in the pilot study; on average, participants talked to 2.33 people (SD = 0.637), ranging from 0 to 8 people. Since the pilot study was conducted with students from the same grade and school as the main study (but different classes), we considered the pre-experimental sample to be homogeneous with the formal sample. In addition, since they share a classroom, the transmitters have the potential to directly affect other groups members though both verbal and non-verbal communication, and there are many potential receivers. We also informed all participants that they could feel free to communicate with each other for a maximum of ten minutes after the transmitters had read the emotion-inducing material and completed their self-report. We did not emphasize the direction of the transmission, and the transmission was unrestricted. We have included some additional information in our Procedure section for clarity (starting on page 8, line 141).

5. In Results section line 206-207, t (56), t (55). Why the df of the two paired sample t-test is different?

Response: Thank you for noting this error. We have corrected it (page 10, line 188).

6. In line 133-134, the author claimed, “The pilot study showed that this material can induce negative collective emotion of the participants.” However, there is no data support this. It would be great if the authors can include their data from the pilot study to back their claim.

Response: We appreciate your feedback. For brevity, we decided not to include much information about the pilot study. Regardless, the manipulation check showed the material to be valid. Please note some additional information about our pilot study below:

In our pilot study, 40 participants (age range = 11 to 13 years) were enrolled in the pre-experiment. We used four materials to elicit emotions. All participants read all four materials and rated their emotions on the Positive Affect and Negative Affect Scale (five-point Likert scale). The results are shown in Supplemental Figure 1, which indicates that the material about movable type printing better evoked negative emotions than did the other materials (F(3, 111) = 5.837, p < .01).

Supplemental Figure 1. Participants’ mean emotional intensity among four emotion-eliciting materials

7. In line 179 after deleting the missing data, what do you mean deleting missing data? I suppose it should be excluding subject who has missing data. What kind of data were missing? How many?

Response: Thank you for your comment. Because four receivers provided less than 50% of all the response items, we deleted their data. We added this fact to our Data analysis section (page 9, lines 160–161).

8. Regarding the subject, are the subject middle school student or high school student? In the Abstract it says middle school, however in other places it says high school student. Need to be accurate and consistent.

Response: Thank you for noting this error. All participants were middle school students; thus, we revised the error (page 19, line 345).

9. The figure legend should put under nether each figure and it is better to put figure within the main text so that it is easier to read.

Response: We originally provided the figures in separate files owing to PLoS One’s guidelines. However, per your wishes, we have now included all four figures in the main text.

10. There are some text formatting issues in line 142-143 (Cronbach’sα). Please correct.

Response: Thank you for noting this error. We have revised this throughout.

Reviewer #2: COMMENTS

This study aimed to explore how emotions are transmitted from some members to the whole group in a face-to-face environment. Through a social event, 158 middle school students were induced to feel anger and disgust. The present study randomly assigned 1/3 of them as senders and others as receivers. Transmitters shared their feelings with the receivers, and the receivers then communicate with other group members. The results show that negative collective emotions are transmitted through the flow from high arousal members to low arousal members. It is converged through the role of emotional contagion. It is accumulated through the role of an emotional cycle, and in the process, feedback strengthens the intensity of emotion. This study shows that the mode of collective emotional transmission consists of three parts: emotional diffusion, contagion and accumulation. This model helps to understand the inherent characteristics of collective emotion transmission, enriches the study of collective emotion, and provides a theoretical reference for the monitoring and management of group events in the future.

Although this study is interesting, the following issues need to be addressed before it is accepted and published.

1. There are some typos and Chinglish expression in the whole manuscript. Please ask native speaker to help modify the language. For example, in line 54, “Collective emotqqions” is a very obvious spelling mistake. In line 142, “with Cronbach’sαof .804” is also a typo. There are still a lot of problems like this, please revise them carefully. In line 211 to 212, “Negative and positive emotion mean, standard deviation, and coefficient of variance before and after collective emotion transmission” is a very typical Chinglish expression. There are many similar expressions in the text, please modify them one by one.

Response: Thank you for your feedback. We have proofread our article thoroughly and sought the services of a professional English-language editing company.

2. “Much research has been performed on cyber collective emotions.” There is a lack of references here. Please add the related references.

Response: Thank you for pointing this out. We have added two relevant references: [4,5] (page 3, line 40).

3. The structure of the introduction is a little loose. In this part, the present study puts forward relevant hypotheses based on previous studies. But the two hypotheses are separated by several paragraphs, and at the end of reading, you will forget the previous hypothesis. It is suggested that a paragraph should be added at the end of the introduction to explain the purpose of this study and the scientific problems to be solved, which will be clearer.

Response: Thank you for your comment. We have added a paragraph at the end of the Introduction to enhance the flow of the paper and to clarify our hypotheses.

4. With regard to the collective emotion induction, I am wondering whether it is beyond their cognition to ask middle school students between the ages of 12 and 14 to rate events with complex political and historical backgrounds?

Response: We completely agree with your comments, and we considered students’ cognitive ability. The art of movable type printing is one of the four major inventions of ancient China, which is common knowledge among middle school students and even elementary school students in China. If any other country officially claimed that movable type printing was an invention of their country, it would likely anger or disgust Chinese individuals. Further, our manipulation check confirmed the validity of this material.

5. In the part of data analysis, what are the criteria and basis for deleting data? This paper does not elaborate on this in detail.

Response: Thank you for your comments. Because four receivers provided less than 50% of all the response items, we deleted their data. We have added this explanation in the Data analysis section (page 9, lines 160–161).

6. Negative emotions include only anger and disgust, while positive emotions include happiness, excitement and satisfaction. Do you consider the problem of quantity mismatch? In addition, are there corresponding criteria and references for dividing positive and negative emotions?

Response: We completely agree with your comments. We used eight emotion words to measure students’ emotional states: sad, happy, angry, disgusted, satisfied, surprised, excited, and calm. Per the results of the factor analysis, angry and disgusted formed the negative dimension, while happy, satisfied, and excited formed the positive emotion dimension. Ideally, the two would have been matched in the number of states.

7. The discussion section is not enough to discuss in detail whether the results are consistent or inconsistent with previous studies, and the relevant explanations. After a brief summary of the research results, the study gives the collective emotion transmission model, and then detailed the current results, but did not explain the relationship between this model and the results of this study. What I am very curious about is whether the results of this study support this model, or do not support this model, or can it be modified? I hope to see the corresponding discussion in the revised draft.

Response: We appreciate your comments and have revised the Discussion accordingly. Specifically, we have included a summary of the results, more thoroughly interpreted the meaning of the results, and compared our findings to those of several other theoretical and empirical studies (please see our revised Discussion starting on page 14, line 254).

6. PLOS authors have the option to publish the peer review history of their article (what does this mean?). If published, this will include your full peer review and any attached files.

Do you want your identity to be public for this peer review? For information about this choice, including consent withdrawal, please see our Privacy Policy.

Reviewer #1: No

Reviewer #2: No

---

## [Decision Letter · Decision Letter 1]

17 Jul 2020

Exploring collective emotion transmission in face-to-face interactions

PONE-D-20-06106R1

Dear Dr. Zheng,

We’re pleased to inform you that your manuscript has been judged scientifically suitable for publication and will be formally accepted for publication once it meets all outstanding technical requirements.

Kind regards,

Zezhi Li, Ph.D., M.D.

Academic Editor

PLOS ONE

Additional Editor Comments (optional):

Reviewers' comments:

Reviewer's Responses to Questions

**Comments to the Author**

1. If the authors have adequately addressed your comments raised in a previous round of review and you feel that this manuscript is now acceptable for publication, you may indicate that here to bypass the “Comments to the Author” section, enter your conflict of interest statement in the “Confidential to Editor” section, and submit your "Accept" recommendation.

Reviewer #1: All comments have been addressed

Reviewer #2: All comments have been addressed

2. Is the manuscript technically sound, and do the data support the conclusions?

Reviewer #1: Yes

Reviewer #2: Yes

3. Has the statistical analysis been performed appropriately and rigorously? 

Reviewer #1: Yes

Reviewer #2: Yes

4. Have the authors made all data underlying the findings in their manuscript fully available?

Reviewer #1: Yes

Reviewer #2: No

5. Is the manuscript presented in an intelligible fashion and written in standard English?

Reviewer #1: Yes

Reviewer #2: Yes

6. Review Comments to the Author

Reviewer #1: The authors have satisfactorily responded to all my questions and made the necessary changes to the manuscript. The revised version of the manuscript appears to be good.

Reviewer #2: (No Response)

7. PLOS authors have the option to publish the peer review history of their article (what does this mean?). If published, this will include your full peer review and any attached files.

Reviewer #1: No

Reviewer #2: No

---

## [Editor Report · Acceptance letter]

27 Jul 2020

PONE-D-20-06106R1 

Exploring collective emotion transmission in face-to-face interactions 

Dear Dr. Zheng:

I'm pleased to inform you that your manuscript has been deemed suitable for publication in PLOS ONE. Congratulations! Your manuscript is now with our production department. 

Kind regards, 

on behalf of

Dr. Zezhi Li 

Academic Editor

PLOS ONE